# Pathophysiological Roles of Stress-Activated Protein Kinases in Pulmonary Fibrosis

**DOI:** 10.3390/ijms22116041

**Published:** 2021-06-03

**Authors:** Yoshitoshi Kasuya, Jun-Dal Kim, Masahiko Hatano, Koichiro Tatsumi, Shuichi Matsuda

**Affiliations:** 1Department of Biomedical Science, Graduate School of Medicine, Chiba University, Chiba 260-8670, Japan; hatanom@faculty.chiba-u.jp (M.H.); elliptical-full.shuichi@nifty.com (S.M.); 2Department of Biochemistry and Molecular Pharmacology, Graduate School of Medicine, Chiba University, Chiba 260-8670, Japan; 3Department of Research and Development, Institute of Natural Medicine (INM), University of Toyama, Toyama 930-0194, Japan; jdkim@inm.u-toyama.ac.jp; 4Department of Respirology, Graduate School of Medicine, Chiba University, Chiba 260-8670, Japan; tatsumi@faculty.chiba-u.jp

**Keywords:** stress-activated protein kinases, c-Jun NH2-terminal kinase, p38 MAPK, idiopathic pulmonary fibrosis

## Abstract

Idiopathic pulmonary fibrosis (IPF) is one of the most symptomatic progressive fibrotic lung diseases, in which patients have an extremely poor prognosis. Therefore, understanding the precise molecular mechanisms underlying pulmonary fibrosis is necessary for the development of new therapeutic options. Stress-activated protein kinases (SAPKs), c-Jun N-terminal kinase (JNK), and p38 mitogen-activated protein kinase (p38) are ubiquitously expressed in various types of cells and activated in response to cellular environmental stresses, including inflammatory and apoptotic stimuli. Type II alveolar epithelial cells, fibroblasts, and macrophages are known to participate in the progression of pulmonary fibrosis. SAPKs can control fibrogenesis by regulating the cellular processes and molecular functions in various types of lung cells (including cells of the epithelium, interstitial connective tissue, blood vessels, and hematopoietic and lymphoid tissue), all aspects of which remain to be elucidated. We recently reported that the stepwise elevation of intrinsic p38 signaling in the lungs is correlated with a worsening severity of bleomycin-induced fibrosis, indicating an importance of this pathway in the progression of pulmonary fibrosis. In addition, a transcriptome analysis of RNA-sequencing data from this unique model demonstrated that several lines of mechanisms are involved in the pathogenesis of pulmonary fibrosis, which provides a basis for further studies. Here, we review the accumulating evidence for the spatial and temporal roles of SAPKs in pulmonary fibrosis.

## 1. Introduction

The stress-activated protein kinase (SAPK) group of mitogen-activated protein kinases (MAPKs) are Ser/Thr kinases and include two members of the c-Jun NH2-terminal kinase (JNK) and p38 MAPK families, which are activated in response to environmental stresses such as UV irradiation, oxidative stress, osmolarity shock, inflammatory stimuli by Toll-like receptor (TLR) ligands and cytokines, withdrawal of trophic factors, chemotherapeutic drugs, and others [1,2]. The JNK family, encoded by three separate but closely related genes, produces three members of JNK1 (*MAPK8*), JNK2 (*MAPK9*), and JNK3 (*MAPK10*). While JNK1 and JNK2 are expressed in various types of cells in adult tissues, the expression of JNK3 is limited and observed in the brain, heart, and testis. The alternative splicing of *Jnk* transcripts results in 10 different JNK isoforms, including four JNK1 isoforms (JNK1α1, JNK1β1, JNK1α2, and JNK1β2); four JNK2 isoforms (JNK2α1, JNK2β1, JNK2α2, and JNK2β2); and two JNK3 isoforms (JNK3α1 and JNK3α2) [3]. These isoforms are divided into two groups based on differences in molecular weight: JNK1α1, JNK1β1, JNK2α1, JNK2β1, and JNK3α1 have a molecular weight of 46 kDa, and JNK1α2, JNK1β2, JNK2α2, JNK2β2, and JNK3α2 have a molecular weight of 54 kDa, with an extended C-terminus. How these isoforms contribute to the overall JNK activity is of interest but needs to be elucidated. JNK1 was originally identified as a specific kinase for c-Jun, which is a key component of activating protein-1, a transcription factor. Subsequent analyses have revealed that JNKs regulate their various substrates, including transcription factors (β-catenin, Elk1/3/4, FOXO3/4, Jun dimerization protein (JDP) 2, JunD, Myc, neural retina leucine zipper, nuclear factor of activated T cells (NFATc) 2, nuclear receptor 4A1, p53, retinoic acid receptor-α (RARA), Sirtuin, STAT1/3, Smad2/3, etc.); kinases (Akt, mammalian sterile 20-like kinase (Mst) 1, p70 ribosomal S6 kinase (p70S6K), etc.); phosphatases (cell division cycle (Cdc) 25B/C, dual-specificity phosphatase (DUSP) 8, and protein phosphatase 1J); cytoskeletal proteins (smoothelin-like protein 2, superior cervical ganglion 10, Tau, doublecortin, WD repeat domain 62, etc.); mitochondrial proteins (myeloid cell leukemia factor 1, Bcl2-associated agonist of cell death, Bcl2-associated X protein (Bax), BH3-interacting domain death agonist (Bid), BH3-only BCL-2-interacting mediator of cell death (Bim), etc.); and other various regulatory proteins [4]. These findings suggest that JNKs may trigger multifunctional signaling cascades. The activation of MAPKs is achieved through three layers of sequentially activated kinases: MAP3K (MAPKKK), MAP2K (MAPKK), and MAPK, which are recognized as a canonical MAPK phosphorylation cascade. JNKs are activated by dual phosphorylation of the TPY motif within their activation loop by two upstream MAP2Ks: MKK4 and MKK7, which are activated by various MAP3Ks, apoptosis signal-regulating kinase 1 (ASK1), dual leucine zipper bearing kinase (DLK), MEKKs, mixed lineage kinases (MLKs), TGF-β-activated kinase 1 (TAK1), thousand-and-one amino acid kinase 2 (TAO2), and TNF receptor-associated factor 2 and NCK-interacting protein kinase (TNIK) [5].

Four mammalian members of p38 (α, β2 (a functional splicing variant of β), γ, and δ) are encoded by different genes: p38α (*MAPK14*), p38β (*MAPK11*), p38γ (*MAPK12*), and p38δ (*MAPK13*). p38α and –β2 are expressed ubiquitously in adult tissues, whereas the predominant expression of p38γ (also known as ERK6 or SAPK3) is observed in skeletal muscle, and p38δ (also called SAPK4) has relatively high levels of expression in the endocrine glands, testes, pancreas, small intestine, kidneys, and lungs [6,7,8]. The downstream substrates of p38 MAPKs include a diverse assortment of protein kinases (MAPK-activated protein kinase (MK) 2/3/5, MAPK-interacting serine/threonine-protein kinase (MNK) 1/2, mitogen- and stress-activated kinase (MSK) 1/2, p21-activated kinase 6, glycogen synthase kinase 3β, phosphatidylinositol 5 phosphate 4-kinase, etc.); transcription factors (ATF2 and CCAAT/enhancer-binding proteins (C/EBPs), CCAAT/enhancer-binding protein homologous protein (CHOP), Elk-1, estrogen receptor (ER) α, Fos, FOXO3a, glucocorticoid receptor, NFATc1/4, p53, paired box 6, peroxisome proliferator-activated receptor (PPAR) α, Smad3, Snail, STAT1/4, Twist1, X-box-binding protein (XBP)-1, etc.); transcriptional coregulators (CREB-regulated transcription coactivator 2, HMG-box transcription factor 1, PPARγ co-activator 1α (PGC-1α), ring finger protein 2, etc.); and other regulatory proteins (Cdc25A/B, cyclin D1/3, Bax, BinEL, Drosha, far upstream element-binding protein 2/3, epidermal growth factor receptor (EGFR), fibroblast growth factor receptor (FGFR) 1, phospholipase A2 (PLA2), TAK1-binding protein (TAB) 1/3, etc.) [9]. Hence, activated p38 MAPKs mobilize a variety of cellular responses. p38 MAPKs are activated by the dual phosphorylation of the TGY motif within their activation loop by two upstream MAP2Ks: MKK3 and MKK6, which are activated by various MAP3Ks, ASK1, MEKKs, MLKs, TAK1, TAO2, and tumor progression locus 2 (TPL2). Although the two SAPK pathways are not redundant, the JNK and p38 pathways share a number of upstream MAP3Ks. Consequently, certain cellular inputs occasionally activate both JNK and p38 in cells. However, whether the two SAPKs function cooperatively or antagonistically should be evaluated by the pharmacological or genetic inhibition of each SAPK [10,11].

As a new p38 activation pathway, the finding that TAB1 selectively binds to p38α and upregulates its autophosphorylation has been reported [12]. In contrast, p38α is capable of phosphorylating TAB1, which leads to the inactivation of TAK1 as a feedback control [13]. TAK1, a MAPK3Ks originally identified as a protein kinase that is activated by TGF-β and bone morphogenic proteins, has been subsequently demonstrated to be activated by various proinflammatory agents such as TNF-α, IL-1β, TLR ligands, and T-cell receptor and B-cell receptor antigens [14]. TAK1 forms the TAB1–TAK1–TAB2 or TAB1–TAK1–TAB3 complex, which is required for autophosphorylation-induced TAK1 activation. TAK1 can activate downstream signaling molecules such as the inhibitor of nuclear factor *κ*B (NF*κ*B) kinase β and MAP2K, including MKK3, MKK4, MKK6, and MKK7, resulting in the activation of NF*κ*B and SAPKs (JNK and p38) [15,16]. As mentioned below, TGF-β1 is known to be a key profibrotic cytokine in the pathogenesis of both idiopathic pulmonary fibrosis (IPF) and various animal models of lung fibrosis [17,18]. Therefore, TAK1, the downstream molecule of TGF-β1 leading to SAPKs activation, attracts attention [19]. In combination with the crosstalk between the JNK and p38 signaling pathways mentioned in the last section, the activation pathways of SAPKs are shown in Figure 1.

Lung fibrosis is recognized as an end-stage pathological development of the existing lung diseases caused by chronic inflammation, infection, autoimmunity, and idiopathy. Progressive pulmonary fibrosis has received considerable attention in recent years, because it causes respiratory failure and an increased risk of death. Most patients with IPF and some patients with nonspecific interstitial pneumonia develop a progressive fibrosing phenotype. In addition, interstitial pneumonia related with collagen tissue disease and chronic hypersensitivity pneumonia exhibit similar clinical course and pathological findings, although they are triggered by an exaggerated immune response and persistent exposure of an inhaled antigen. Various combinations of repeated alveolar, or endothelial cell injury, immune activation, and inflammation, initiate the following fibrotic process. Myofibroblasts proliferation and extracellular matrix deposition promoted by profibrotic cytokines results in lung tissue remodeling. These are common cascades in progressive pulmonary fibrosis [20]. Patients with IPF show a poor prognosis correlated with the extent of fibrotic focus formation (median survival of 3–5 years) and have an increased risk of pulmonary hypertension and lung cancer, making IPF one of the most devastating lung diseases [21,22]. While nintedanib and pirfenidone (PFD) have been approved as the current pharmacological therapies for IPF by the Food and Drug Administration (FDA), neither can improve the survival of patients with IPF [23]. Hence, new beneficial strategies that enable patients with IPF to survive longer and improve their quality of life are needed. In addressing these unmet needs, the animal models of experimental lung fibrosis play prominent roles, although they do not recapitulate IPF [24].

Accumulated reliable evidence from the analyses of samples from patients with IPF and animal models of lung fibrosis provide a conceptual model of IPF development driven by a cascade of injuries as follows: (1) damage of the epithelial cells of various pro-aging factors; (2) damage to the accumulation-promoted cellular senescence-associated secretory phenotype, resulting in the release of a broad repertoire of cytokines, chemokines, growth factors, and matrix remodeling proteases; (3) innate and adaptive inflammation; (4) the establishment of myofibroblasts from various types of cells, including fibroblasts (known as fibroblast-to-myofibroblast differentiation, FMD), epithelial cells (known as epithelial–mesenchymal transition, EMT), endothelial cells (known as endothelial–mesenchymal transition, EndoMT), and so on; and (5) excessive scar formation associated with ECM deposition, resulting in dyspnea [21,25,26,27,28,29,30].

Since the contribution of p38, a member of SAPKs, to bleomycin (BLM)-induced lung fibrosis was first shown, 20 years have passed [31]. Many researchers have investigated the spatial and temporal roles of JNK and p38 in pulmonary fibrogenesis to better understand the mechanisms underlying pulmonary fibrosis for the development of new therapeutic options. In this review, we provide an outline of previous reports and refer to future prospects.

## 2. Involvement of JNK in the Pathogenesis of Pulmonary Fibrosis

### 2.1. Activation of JNK in Epithelial Cells Leading to Cell Death

The cell death of alveolar epithelial cells (AECs) is believed to trigger pulmonary fibrogenesis [32]. Regulated cell death (RCD) is a molecularly oriented definition of terms such as intrinsic apoptosis, extrinsic apoptosis, mitochondrial permeability transition (MPT)-driven necrosis, necroptosis, ferroptosis, pyroptosis, parthanatos, NETotic cell death, entotic cell death, lysosome-dependent cell death, autophagy-dependent cell death, immunogenic cell death, cellular senescence, and mitotic catastrophe [33]. Among them, intrinsic apoptosis is a form of RCD induced by microenvironmental perturbations such as reactive oxygen species (ROS) overload, endoplasmic reticulum (ER) stress, DNA double-stranded breaks, and so on [34,35,36]. Meanwhile, extrinsic apoptosis occurs secondary to extracellular perturbations that are mostly driven by two types of receptors: dependence receptors and death receptors [33,37]. Death receptors are not limited to, but include, TNF receptor superfamily members (TNFR1, Fas—known as TNFR6, TRAILR1, and TRAILR2) [38]. These intracellular and extracellular microenvironmental perturbations are often observed in the progression of lung fibrosis and are related to SAPK signaling. For instance, the primary action of BLM, most frequently used in a murine model of experimental lung fibrosis, is double-stranded DNA cleavage [24,39]. Therefore, focusing on SAPK-mediated intrinsic/extrinsic apoptosis is reasonable for evaluating lung fibrogenesis. In addition, the involvement of cell death, different from apoptosis in lung fibrogenesis, is a recent topic. It has been reported that the genetic downregulation of glutathione peroxidase 4 (GPX4) expression exacerbates BLM-induced lung fibrosis. Considering that GPX4 evokes a countereffect on phospholipid (PL) peroxidation, ferroptosis, defined as a PL peroxidation-induced sudden cell death, may be involved in the development of lung fibrosis [40]. Although whether SAPKs mediate ferroptosis-associated lung fibrosis is still unclear, the MLK3/JNK signaling axis at least partly contributes to ferroptosis and pyroptosis in the pathogenesis of myocardial fibrosis [41].

It has been demonstrated that the apoptosis of AECs in response to cell death inducers such as BLM, the Fas ligand, and TNF-α requires the autocrine synthesis of angiotensin II (ANG II) by AECs [42,43,44]. Notably, the expression of ANG II and its specific G protein-coupled receptors (GPCRs, AT1, and AT2) increases in lung tissue from patients with IPF, the major sources of which are AECs and myofibroblasts [45,46]. ANG II, the multiple bioactive octapeptide, is produced through the enzymatic cascade: the decapeptide ANG I is produced from the NH2-terminal end of angiotensinogen by renin or other aspartyl proteases; ANG II is then formed by the cleavage of two amino acid residues from the COOH-terminal end of ANG I by angiotensin-converting enzyme (ACE), chymase, or other peptidases. In addition, ACE-2 is known to degrade ANG II by removing the COOH-terminal phenylalanine (Phe) residue to yield ANG1-7. Although ANG II plays various physiological and pathophysiological roles through its binding to AT1 and AT2, the well-known actions of Ang II are mainly mediated through the AT1 receptor [47]. The binding of ANG II to AT1 triggers a variety of kinase signaling cascades, such as SAPKs/MAPKs, Rho kinases, glycogen synthase kinase, receptor tyrosine kinases, and nonreceptor tyrosine kinases [48]. On the other hand, ANG1-7 has its own GPCR Mas, the product of the oncogene Mas [49]. As ANG1–7 binds to Mas and counteracts the resulting effects of the ANG II/AT1 signaling axis, how the ACE-2/ANG1-7/Mas signaling axis behaves under some disease conditions is of interest [50]. In adult human lungs, ACE-2 is expressed in type I and II AECs, bronchial epithelial cells, and the blood vessel system [51]. Uhal et al. showed that the levels of ACE-2 mRNA, protein, and enzymatic activity are severely reduced in lung tissues from patients with IPF [52]. In the lungs from control subjects, a high level of ACE-2 expression was detected in type II AECs in a quiescent state [53]. In addition, the treatment of type II AECs with ANG II or BLM induces caspase activation, nuclear fragmentation, and JNK phosphorylation in the cells, which is inhibited by the pretreatment of the cells with ANG1-7 or a JNK inhibitor (Inh) (SP600125). Furthermore, the inhibitory effect of ANG1-7 on ANG II/AT1/JNK-signaling axis-associated apoptosis was abrogated by A799, a Mas receptor blocker [54]. Therefore, the imbalance of the ANG II/AT1/JNK axis vs. the ACE-2/ANG1-7/Mas axis leads to the apoptosis of AECs, which may be an important step in lung fibrogenesis.

Recently, it was reported that a homozygous missense mutation in the surfactant protein A1 gene (*SFTPA1*) caused IPF in a consanguineous Japanese family [55]. The report provided informative evidence: (1) a homozygous missense mutation in *SFTPA1* (T622C) associated with an amino acid change in the carbohydrate recognition domain of SP-A1 (Y208H) was revealed by a homozygosity mapping, (2) mice carrying the mutation in *Sftpa1* (Sftpa1-knock-in (KI) mice) spontaneously developed pulmonary fibrosis at 20 weeks (W) of age with further deterioration with aging, (3) the cell death of type II AECs in Sftpa1-KI mice was prominent at 30 W, the type of which was necroptosis but not apoptosis because of the sensitivity to receptor-interacting serine/threonine kinase 3 (RIPK3) deficiency, (4) ER stress before the cell death of type II AECs activated JNK, which, in turn, upregulated the expression of RIPK3, and (5) JNK Inh (SP600125) or the deletion of *Ripk3* or mixed lineage kinase-like (*MLKL*) gene ameliorated lung fibrosis in Sftpa1-KI mice. Human pulmonary surfactant proteins (SP-A1 and -A2, SP-B, SP-C, and SP-D) are synthesized and secreted by type II AECs and contribute to the maintenance of pulmonary tissue stability. SP-B and SP-C are hydrophobic and important for normal lung function.SP-A1, SP-A2, and SP-D are hydrophilic and are primarily involved in host defense [56,57]. SP-A1 (SFTPA1) forms a complex with SP-A2 (SFTPA2), which is secreted into the alveolar space. The accumulation of misfolded SP variants induces ER stress in type II AECs [58,59]. In addition, in the case of Sftpa1-KI mice, the homologous missense mutation resulted in the failure of SP-A secretion [55]. Furthermore, the cell death triggered by misfolded protein-associated ER stress is at least partly mediated through JNK or p38, which is activated downstream the inositol-requiring enzyme (IRE)-1α [60]. In necroptosis, active RIPK3 phosphorylates the pseudokinase MLKL, which drives its oligomerization and translocation to the membrane. MLKL, which forms a channel-like structure, binds specific phosphatidylinositol phosphate species by a rollover mechanism and, hence, triggers plasma membrane permeabilization [33]. The fact that ER stress mobilizes the JNK/RIPK3-signaling axis function as an initiator of lung fibrosis through the necroptosis of type II AECs is of interest. Future analyses are required to determine whether this signal is universally mobilized by other ER stress inducers.

#### Activation of JNK in Epithelial Cells Leading to EMT

EMT is well-known as one of the established routes for myofibroblasts [29]. The mortality of patients with IPF is correlated with the extent of fibrotic focus formation, which results from the myofibroblast-derived abnormal and excessive accumulation of ECM components, including collagen, fibronectin, and elastin [22]. TGF-β1 potently induces EMT with changes in certain markers, such as an upregulation of myofibroblast markers (α-SMA, collagen, and fibronectin) and mesenchymal markers (vimentin and neural cadherin), in proportion to a downregulation of epithelial markers such as E-cadherin [61]. Janssen-Heininger’s group demonstrated the involvement of JNK1-mediated EMT in lung fibrosis as follows: (1) using mouse tracheal epithelial cells from wild-type (WT), JNK1^−/−^, or JNK2^−/−^ mice, the finding that JNK1 but not JNK2 is required for TGF-β1-induced EMT was shown [62]; (2) accumulated subepithelial collagen deposition associated with the induction of profibrotic molecules, including TGF-β1, in response to a sensitization/challenge with OVA was less in JNK1^−/−^ mice than in WT mice. Although JNK1^−/−^ mice showed no impact on the recruitment of inflammatory cells observed after the administration of BLM, collagen deposition in a BLM-induced fibrotic lung was inhibited in JNK1^−/−^ mice compared to WT mice [63]; (3) conditional ablation of *Jnk1* in bronchiolar epithelial cells and type II AECs caused a strong resistance to BLM- or adenovirus expressing active TGF-β1 (AdTGFβ1)-induced fibrosis; and (4) the delayed epithelial specific ablation of *Jnk1* by the tetracycline operon driving the CRE recombinase system prevented AdTGFβ1-induced fibrotic remodeling, suggesting that the site-specific inhibition of JNK1 reverses established lung fibrosis [64]. It is generally known that most patients with IPF presenting with subjective symptoms have already progressed to the early fibrotic stage with reduced forced vital capacity [65]. Hence, the spatial and temporal inhibition of JNK1 may be a potential therapeutic option.

### 2.2. Activation of JNK in Lung Fibroblasts Leading to Myofibroblastic Phenotypes

The elevated secretion of endothelin (ET)-1 was observed in fibroblasts isolated from patients with a fibrotic lung of scleroderma but not in normal subjects [66]. Shi-Wen et al. investigated the pathological significance of ET-1 in lung fibroblasts from patients with fibrosing alveolitis associated with systemic sclerosis (FASSc): (1) TGF-β1 induced the transcriptional activation of the *ET-1* gene, resulting in the upregulation of the ET-1 peptide in normal human lung fibroblasts (NHLF), which was inhibited by interfering with the JNK1/c-jun/AP-1 signaling axis but not the ALK5 (activin receptor-like kinase 5, also known as TGF-βR I)/Smad2/3 pathway; (2) TGF-β1-induced upregulation of α-SMA was markedly inhibited by bosentan (nonselective blocker for ENDRA/ENDRB); (3) upregulation of ET-1 mRNA expression and JNK1 activation was observed in FASSc fibroblasts compared to NHLF, which was significantly inhibited by bosentan; and (4) in fibroblasts derived from TAK-1^−/−^ mice, the ET-1-induced JNK1 activation-associated α-SMA expression was much suppressed compared to the WT fibroblasts [66]. Supportive data has demonstrated that the ET-1/ENDRA axis-induced expression of α-SMA and connective tissue growth factor is mediated by the JNK/AP-1 pathway in human lung fibroblasts [67]. Therefore, these findings suggest that the Smad-independent TGF-β1/TAK1/JNK1/AP-1 axis leading to the upregulation of ET-1 expression may be constitutively activated in FASSc fibroblasts and that the autocrine ET-1 loop mediated by JNK1 activity maintains the myofibroblastic phenotype. Bosentan, approved by the FDA, is an orally active drug for the treatment of pulmonary arterial hypertension [68]. Although interventional clinical trials have been conducted with bosentan in patients with IPF, the expanded application of bosentan remains to be elucidated [69,70] (clinicaltrials.gov, registration number NCT03074149).

Caveolae, observed as omega-shaped invaginations of the plasma membrane, are characterized by the existence of an integral membrane protein family, caveolin (Cav)-1, -2, and -3. Cav-1 is the principal structural component of caveolae and is necessary for the appearance of caveolae. Cav-1 can bind to various classes of signaling molecules, including MAPKs, small G proteins, receptor tyrosine kinases, nonreceptor tyrosine kinases, endothelial NO synthase, and glucocorticoid receptors, resulting in negative regulators of signal transduction [71,72]. Hence, the downregulation of Cav-1 expression may promote various pathological microenvironments. Cav^−/−^ mice display vascular system dysfunction and pronounced thickening of the lung alveolar septa due to the uncontrolled proliferation of endothelial cells and fibrosis [73]. Wang et al. investigated the relationship between the changes in Cav-1 expression and profibrotic signal in fibroblasts: (1) in the lungs of patients with IPF, the marked downregulation of Cav-1 at the mRNA and protein levels attributed to AECs and fibroblasts, (2) the gene transfer of Cav-1 by adenovirus suppressed BLM-induced lung fibrosis, (3) in human lung fibroblasts, TGF-β1 induced FMD, associated with the downregulation of Cav-1 and upregulation of ECM proteins, which was reversed by the overexpression of Cav-1, (4) in human lung fibroblasts, TGF-β1 induced the Smad2 activation-associated nuclear translocation of Smad2/3 and phosphorylation of extracellular signal-regulated kinase (ERK) and JNK, all of which were inhibited by the overexpression of Cav-1, and (5) TGF-β1-activated Smad signaling was markedly suppressed in fibroblasts from JNK1^−/−^ mice compared to those from WT mice, indicating that the Cav-1-modulated JNK1 activity exists upstream of Smad. This was supported by the finding that Smads (Smad2/3) are relevant substrates of JNK [4]. (6) Additionally, in fibrotic lung lesions from patients with IPF, an extensive signal of phospho-JNK was detected [74]. Therefore, there is a “tug of war” between Cav-1 and JNK1, at least in the establishment of TGF-β1-promoted myofibroblasts under fibrogenesis.

The overview of the JNK-associated lung fibrogenesis introduced here is shown in Figure 2. Among JNKs, the expression of JNK1 and JNK2 is observed in the lung parenchymal cells. Likewise, SP600125, a commonly used JNK inhibitor, is effective with JNK1, as well as JNK2 [75]. Hence, isoform-specific gene ablation or knockdown is required to estimate which JNK isoform plays a more crucial role in lung fibrogenesis. In fact, experimental models and several tissues where the actions of JNK1 and JNK2 are cooperative or synergistic have been reported [4]. Therefore, as for the findings in which a clear difference between JNK1 and JNK2 in terms of their contributions to fibrogenesis was not established, the JNKs were defined as JNK1/2 (Figure 2).

## 3. Involvement of p38 in the Pathogenesis of Pulmonary Fibrosis

### 3.1. Activation of p38 in Epithelial Cells Leading to Injury

The complement system is an integral part of the adaptive and innate immunity that promotes the host defense against pathogenic challenges. Among more than 30 complement proteins, C3a and C5a (anaphylatoxins), generated from the cleavage of C3 and C5, respectively, are involved in various inflammatory responses. Complement inhibitory proteins (CIPs) are responsible for confining complement activation to appropriate contexts to prevent errant complement activation in healthy hosts [76]. The cell membrane-bound CIPs, cluster of differentiation 46 and 55 (CD46 and CD55), are ubiquitously expressed in the upper and lower respiratory tracts of normal humans [77]. Gu et al. investigated the crosstalk between the complement, TGF-β1, and p38 in regulating CIP expression in the airway epithelium: (1) the expression level of CD46 and CD55 in lung tissue homogenates from patients with IPF was much lower compared to that of normal subjects, which resulted from their downregulation, especially in AECs and airway epithelia. Similarly, the systemic and local levels of the anaphylatoxins C3a and C5a were higher in patients with IPF than in normal subjects; (2) in normal primary human small airway epithelial cells (SAECs), TGF-β1 downregulated the expression of CD46, CD55, and E-cadherin and upregulated the expression of Snail, which is recognized as a marker of EMT because of its transcriptional repressor activity to *CDH1* (E-cadherin gene). C3a and C5a also downregulated the expression of CD46 and CD55, in which a cleaved/active form of poly (ADP-ribose) polymerase, a marker of apoptosis-mediated cell injury, increased; (3) the TGF-β1-induced downregulation of CD46, CD55, and E-cadherin was blocked by a p38 Inh (SB203580) but not by a JNK Inh or an ERK Inh; (4) in SAECs, C3a and C5a sequentially induced p38 activation, a decrease in the expression of CD46 and CD55, upregulation of Snail expression, and downregulation of E-cadherin expression; (5) the C3a/C5a-induced changes in the expression of CD46, CD55, and Snail were p38 Inh-sensitive; and (6) while TGF-β1 induced the expression of the C3a and C5a receptors, both C3a and C5a downregulated Smad7, a negative regulator of the TGF-β signal in SAECs [78]. Hence, the crosstalk between TGF-β1 and the complement system augments epithelial injuries in lung fibrogenesis, where p38 may play a crucial role. Although the p38 inhibitor SB203580 is effective with p38α and β, p38α is shown to phosphorylate Snail and enhance its stability [9,79,80]. Hence, among the p38 isoforms, p38α may be predominantly involved in the crosstalk event between TGF-β1 and the complement system.

#### 3.1.1. Activation of p38 in Lung Fibroblasts Leads to Profibrotic Characters

FMD is well-known as the main route of myofibroblast establishment [81]. It has long been thought that myofibroblasts are recognized as irreversible differentiated cell states. However, the discovery of drugs that resolve established fibrosis by promoting myofibroblast dedifferentiation is becoming active [28,82,83]. Therefore, promoting both the efficient inhibition of FMD and reversal of established myofibroblasts has great potential as a beneficial intervention in patients with IPF. To address this particular proposition, accumulated information regarding the precise characteristics of fibroblasts/myofibroblasts is needed.

Proteoglycans (PGs) are key components of the ECM and consist of a core protein to which multiple sulfated glycosaminoglycan (GAG) side chains are attached. GAGs can interact with growth factors [84], chemokines, and cytokines by virtue of a highly negative charge from the sulfate group, regulating a wide variety of cellular processes [85]. In the biosynthetic reactions of GAGs, xylosyltransferase (XT)-I and glucuronosyltransferase (GlcAT)-I are responsible for initiation and termination, respectively. Chondroitin-4-sulfotransferase (C4ST)-I, which attaches a sulfate group to the 4-O position of an *N*-acetylgalactosamine residue in chondroitin sulfate (CS) chains, is also a key enzyme for GAGs [86]. Venkatesan et al. demonstrated the involvement of p38 in GAG biosynthesis in myofibroblasts: (1) the expression level of XT-I mRNA and C4ST-I mRNA and the secretion amount of CS-GAG increased in myofibroblasts compared to fibroblasts; (2) TGF-β1-induced p38 activation was associated with the upregulation of XT-I, C4ST-I, CS-GAG, and versican (known as the CS-PG core protein), which was blocked by ALK5 Inh or p38 Inh (SB203580); and (3) in AdTGFβ1-induced lung fibrosis model mice, the levels of XT-I, C4ST-I, CS-GAG, and versican were upregulated compared to control mice, all of which were localized in the fibrotic lesions [87]. In addition to the canonical Smad-independent TGF-β1/TAK1/SAPKs (JNK and p38) signaling pathway, the Smad-dependent TGF-β1/ALK5/p38 pathway is also known [88,89]. Hence, in regard to the effect of SB203580, the ALK5/p38α/β signaling axis regulates PG deposition in lung fibrogenesis. To precisely determine ECM deposition in fibrogenesis, focusing not only on fibrous proteins, including collagen, fibronectin, elastin, and laminin but, also, PGs is important, because GAGs and versican are abundant within the myofibroblast core of the fibroblastic focus and promote a variety of fibrogenic cellular functions [90].

#### 3.1.2. Activation of p38 Signal in Lung Fibroblasts Leading to Invasion Activity

Hyaluronan (HA), a GAG biosynthesized by HA synthase 2 (HAS2), stimulates cell invasiveness via interaction with its major receptor, CD44 [90]. Fibroblasts are recognized as one of the major HA-producing cells in the lungs. Noble’s group demonstrated the involvement of MK2, which functions downstream p38 in the invasive capacity of fibroblasts: (1) both the invasive phenotype of fibroblasts and BLM-induced progressive fibrosis were inhibited by the loss of HAS2 or CD44. Similarly, fibroblasts isolated from the lungs of patients with IPF showed an increased invasive capacity compared to those of normal subjects, which were dependent on HAS2 and CD44 [91]; (2) the IPF myofibroblasts exhibiting an invasive phenotype showed a high production of HA associated with HAS2 upregulation, which was blocked by MK2 Inh (MMI-0100). In contrast, HAS2 knockdown reduced MK2 phosphorylation in cells, indicating a feedforward loop between MK2 activation and HAS2 expression; (3) BLM induced collagen deposition-associated lung fibrosis, the upregulation of HAS2 mRNA expression, and an increase in HA production, all of which were sensitive to MMI-0100; and (4) after the BLM instillation, fibroblasts carrying a specific deletion of the *Mk2* gene showed decreased HA production, reduced HAS2 mRNA expression, and suppressed invasion capacity, compared to fibroblasts from the WT littermates [92]. In the activation of the HA/CD44 signaling axis, ALK5 (TGF-βR I) is crucial, because CD44 physically links to ALK5 but not TGF-βR II. HA-activated ALK5 phosphorylates CD44, which enhances its interaction with ankyrin, a scaffolding protein that links membrane-bound proteins to the underlying cytoskeleton, resulting in HA-mediated cell migration [93]. Furthermore, TGF-β1 upregulates the expression of both HAS2 and CD44 [94]. Hence, such a feedforward loop of the MK2-mediated HA/CD44 signaling axis may contribute to lung fibrogenesis by regulating the invasive capacity of myofibroblasts. As for MK2 activation in lung fibroblasts, its regulatory role in the TGF-β1-induced myofibroblastic phenotype has been reported. Notably, the systemic or local administration of MMI-0100 (a cell-permeable synthetic peptide composed of 22 amino acid residues) to mice reverses established lung fibrosis by BLM, suggesting its clinical application [95]. Although MK2 is theoretically a downstream substrate of all p38 isoforms, the best characterized member of the family regulating MK2 is p38α [96]. Furthermore, among the p38 family members, p38α is shown to most potently regulate the cell migration where MK2 activation is required [97]. Therefore, the MK2 activation-associated HA/CD44 signaling axis may be mainly mediated by p38α. To clarify this possibility, CDD-450 (also known as ATI-450) that selectively binds to and inhibits the p38α–MK2 complex is useful [98].

P-Rex1, a PI3K- and Gβγ-regulated guanine nucleotide exchange factor, is known to activate the small G-protein Rac. A recent analysis using the 3D matrix environment system demonstrated that *Rac1*^−/−^ fibroblasts increase the cell–cell adhesion force and reciprocally lose 3D motility [99]. Liang et al. investigated the involvement of P-Rex1 in lung fibrogenesis: (1) *Prex1* deficiency ameliorated BLM-induced lung fibrosis and significantly reduced the mortality rate in mice. Similarly, the BLM-induced expression of profibrotic/proinflammatory cytokines was suppressed in *Prex1*^−/−^ mice compared to WT mice; (2) TGF-β1-induced cellular migration was much slower in *Prex1*^−/−^ fibroblasts than in WT fibroblasts, which was replicated by the treatment of WT fibroblasts with Rac1 Inh. Moreover, TGF-β1-induced Rac1 activation was blocked by a dominant-negative (DN) mutant of P-Rex1 in fibroblasts; and (3) the BLM-induced activation of p38 in the lungs was diminished in *Prex1*^−/−^ mice compared to WT mice. Furthermore, the TGF-β1-induced p38 phosphorylation in fibroblasts was abolished by *Prex1* deficiency, which was replicated by treating WT fibroblasts with Rac1 Inh [100]. Hence, the TGF-β1-activated P-Rex1/Rac1/p38 signaling axis may participate in the motility of fibroblasts, affecting lung fibrogenesis. In this case, which p38 isoform is mobilized remains to be elucidated.

### 3.2. Activation of the p38 Signal in Macrophages Leading to Profibrotic

Monocytes/macrophages comprise a variety of subsets with diverse functions. Macrophages can be classified as two major subtypes, which have been characterized broadly as proinflammatory/cytotoxic M1 macrophages (also known as classically activated macrophages) and anti-inflammatory/wound repair M2 macrophages (also known as alternatively activated macrophages). M2-polarized alveolar macrophages play an important role in the pathogenesis of fibrosing disorders [101]. Considering the progression of lung fibrosis, where the inflammatory and fibrotic stages interlace with one another, evaluating the macrophage M1M2 polarization balance may also be important [102]. Moreover, it has also been demonstrated that atypical monocytes possessing a unique phenotype and cell lineage are activated specifically in lung fibrogenesis [103].

To date, the data regarding how SAPKs control monocytes/macrophages in lung fibrogenesis have been less than those in the cases of the epithelium and fibroblasts. However, Gu et al. provided the possible involvement of p38 in M2 macrophage polarization under the development of lung fibrosis, focusing on mitochondrial calcium uniporter (MCU)-mediated fatty acid oxidation (FAO) in macrophages: (1) MCU expression polarized macrophages to a profibrotic phenotype during asbestos-induced lung fibrogenesis. Likewise, *Mcu*^+/−^ mice showed marked resistance to asbestos-induced lung fibrosis [104]; (2) the elevated expression of MCU was detected in the mitochondria of macrophages from patients with IPF compared to those from normal subjects, which was associated with the changes in Ca^2+^ concentration in the mitochondria. Moreover, a key enzyme for FAO, peroxisome proliferator-activated receptor gamma coactivator 1-alpha (PGC-1α), was upregulated in IPF macrophages; (3) chimeric mice, WT with *Mcu*^+/−^-bone marrow (BM), showed resistance to BLM-induced lung fibrosis. In contrast, chimeric mice, *Mcu*^+/−^ mice with WT-BM, developed lung fibrosis in response to BLM, in which alveolar macrophages showed a drastic increase in the expression of M2 markers (CD206 and MARCO). Furthermore, the silencing of PGC-1α abrogated the profibrotic phenotype; (4) in MH-S cells (a murine alveolar macrophage cell line), the overexpression of MCU and MCU^DN^ (a DN mutant of MCU), respectively, upregulated and downregulated the promoter activity of PGC-1α under control of the p38/ATF-2 signaling axis. Accordingly, the MCU-upregulated PGC-1α promoter activity was augmented by the overexpression of MKK6-CA (a constitutively active mutant of MKK6) and suppressed by the overexpression of p38α-DN; and (5) in THP-1 cells (human monocytic leukemia cell line), MCU-mediated ROS production in the macrophage mitochondria was abolished by the silencing of Rieske, the iron–sulfur protein in complex III, which showed good parallelism with a marked inhibition of the p38/ATF-2-signaling axis and PGC-1α expression [105]. In addition, p38α/β can directly phosphorylate and stabilize PGC-1α, which is associated with the upregulation of its transcriptional activity [9,106]. Hence, the MCU/PGC-1α-regulated M2 polarization of macrophages can be mediated by p38α during lung fibrogenesis.

A recent report regarding the function of forkhead box M1 (FOXM1), one of the FOX family of transcriptional factors in macrophages, by Goda et al. also suggested that p38 activation in macrophages augments lung fibrogenesis: (1) based on the finding that FOXM1 expression changed in macrophages in lung fibrotic lesions from patients with IPF, mice with the deletion of *Foxm1* in myeloid cells (*myFoxm1*^−/−^) were subjected to BLM instillation. Then, exacerbated lung fibrosis was found, indicating the antifibrotic function of FOXM1 in macrophages; (2) upregulated profibrotic genes, downregulated dual-specificity phosphatase 1 (DUSP1, also known as MAP kinase phosphatase-1), and increased phosphorylation of p38 were observed in macrophages derived from BLM-instilled *myFoxm1*^−/−^ mice, compared to the case of WT mice; (3) the knockdown of FOXM1 resulted in a downregulation of DUSP1-associated p38 activation and an increase in the production of several cytokines in RAW264.7, the culture supernatant of which induced FMD; and (4) the chromatin immunoprecipitation assay and luciferase assay revealed that FOXM1 transcriptionally activated the *Dusp1* promoter. The overexpression of DUSP1 in *Foxm1*^−/−^ macrophages suppressed the upregulation of p38 phosphorylation and cytokine production [107]. Hence, the FOXM1/DUSP1 axis regulates lung fibrogenesis by controlling the p38 activity in macrophages. DUSP1 is a so-called nuclear inhibitor for p38 and JNK [108]. In conjunction with the decrease in expression of DUSP1, the selective activation of p38 but not JNK, remains to be elucidated. In addition, which p38 isoform is predominantly activated is also unknown.

In contrast, mice with a systemic deletion of *Dusp10*, which is an equipotent phosphatase of JNK and p38 (also known as MKP-5), show clear resistance to BLM-induced lung fibrosis [109]. In this situation, macrophages were markedly polarized to the M1 phenotype. However, it is noteworthy that *Dusp10*^−/−^ fibroblasts exhibiting a hyperactivation of p38, but not JNK, did not show TGF-β1-induced FMD indices involving an increase in myofibroblast marker expression, upregulation of invasive capacity, and activation of the Smad pathway. Therefore, the loss of DUSP10-associated p38 activation in fibroblasts may play an antifibrotic role. As mentioned above, the myofibroblastic differentiation and invasion capacity of fibroblasts are at least regulated by p38α. Which p38 isoform is activated in response to the loss of DUSP10 is of interest from the viewpoint of whether the bidirectional effects are elicited among the p38 isoforms in the same cell.

An overview of the p38-associated lung fibrogenesis introduced here is shown in Figure 3. As for the findings in which a clear difference among four p38 isoforms in terms of their contributions to fibrogenesis were established, each isoform name is indicated, and others are simply defined as p38 (Figure 3).

### 3.3. Application of Profibrotic Function of p38 Signal to Identify the Therapeutic Target of IPF

Based on the established notion that p38 positively regulates lung fibrogenesis in most cases, we generated the following transgenic mice: C57BL/6J-hSP-C-M2 flag-p38α DN (p38-DN, dual mutations in mouse p38α: Thr180 to Ala and Tyr182 to Phe) and C57BL/6J-hSP-C-3HA-tag-MKK6 CA (MKK6-CA, dual mutations in human MKK6: Ser207 to Asp and Tyr211 to Asp) TG mice. After confirming that each transgene-derived protein localizes at least in SP-C-positive type II AECs, the two genotypes and WT littermates were subjected to BLM-induced lung fibrosis. We expected that mice with a theoretically different intrinsic activity of the p38 signal (p38-DN < WT < MKK6-CA) in type II AECs would show a stepwise worsening severity of BLM-induced lung fibrosis. If so, the genes that changed in proportion to the three different severity levels might be recognized as more reliable therapeutic targets. According to our expectations, a worsening severity of lung fibrosis was associated with an increased intrinsic p38 activity in the lungs of BLM-instilled mice, which was evaluated using the following indices: histopathological alteration, Ashcroft scale, collagen content, inflammatory cell infiltration, and static lung compliance. To conduct an RNA-seq/transcriptome analysis, the lungs of the three genotypes at 8 days post-installation (dpi) with BLM were subjected to mRNA extraction. At approximately 8 dpi, the lungs showed a transition from the inflammatory to the fibrotic phase and a tendency of decrease in static lung compliance, which was more likely to be correlated with patients with IPF presenting subjective symptoms [65]. Considering the beginning period of pharmacologic intervention for patients with IPF, searching for therapeutic targets at 8 dpi is reasonable. The K-means clustering analysis showed that a cluster of 2722 genes increased along with the stepwise elevation of p38 signaling in BLM-instilled lungs. Among these, 137 differentially expressed genes (DEGs) were identified as commonly upregulated genes (vs. PBS treatment) in the three genotypes. Finally, in comparison with human RNA-seq data providing 475 upregulated DEGs in IPF lung tissue from the Gene Expression Omnibus website (accession ID: GSE52463), four overlapping DEGs were identified: EPH receptor A3 (EPHA3); POU class 2 homeobox associating factor 1 (POU2AF1); SAM domain, SH3 domain and nuclear localization signals 1; and ectodysplasin A2 receptor [110]. Among the four candidates, the importance of EPHA3 and POU2AF1 in the regulation of pulmonary fibrosis has been reported as a topic [111,112]. Therefore, although the direct evidence regarding the involvement of other candidates in lung fibrogenesis remains to be elucidated, our strategy is reliable as an approach to identify the therapeutic targets of IPF.

## 4. Is an Anti-SAPK Strategy Effective Against IPF?

Heat shock protein (HSP)90 is a regulator of cell homeostasis and acts as one of the chaperones assisting the folding and assembly of newly synthesized proteins. The HSP90 family is composed of four members: HSP90α, HSP90β, GRP94, and tumor necrosis factor receptor-related protein 1. Among them, HSP90α and HSP90β are the key members and are known to function as extracellular proteins in addition to intracellular chaperones [113]. Notably, the HSP90α level is upregulated in patients with IPF and shows a good correlation to the disease severity, which strongly suggests that HSP90α is a possible diagnostic biomarker, as well as a therapeutic target, of IPF. Furthermore, intracellular HSP90β (iHSP90β) stabilizes low-density lipoprotein receptor-related protein 1 (LRP1), a receptor for extracellular HSP90α (eHSP90α), thus amplifying the eHSP90α actions, such as the promotion of FMD and maintenance of the myofibroblastic phenotype [114]. As with the case of patients with IPF, the level of eHSP90α in the blood and bronchoalveolar lavage fluid has been shown to increase in a BLM-induced lung fibrosis model. In addition, 1G6-D7, a monoclonal antibody to block the effects of eHSP90α, ameliorates BLM-induced lung fibrosis [115]. The eHSP90α/LRP1 signaling axis activates ERK (one of the MAPKs), Akt, and p38, although the ERK signaling pathway predominantly contributes to the cellular output. Furthermore, iHSP90α has been reported to stabilize TGF-βRs, ALK5, and TGF-βRII, leading to the positive regulation of the TGF-β1 signaling pathway [116]. This finding suggests that there is a possible crosstalk between HSP90α and SAPKs. Hence, whether a concomitant inhibition of HSP90 and SAPK shows synergistic therapeutic effects on lung fibrosis is of interest as a future study.

PFD was originally recognized as a small-molecule p38γ inhibitor that blocks the synthesis of TGF-β [117]. However, the inhibitory effect on p38 has been excluded from the key antifibrotic mechanisms of PFD [118]. Currently, a clinical trial with the SAPK inhibitor for IPF in phase II is in progress, in which an orally active inhibitor of JNK1, CC-90001, is being studied (clinicaltrials.gov, registration number NCT03142191). Unfortunately, the inhibitors of SAPKs that are energetically developed have not met defined endpoints in various clinical trials targeting IPF, as well as inflammatory diseases. Although various situations should be addressed as the reasons, a crosstalk between p38 and JNK cannot be negligible from the viewpoint of signaling. As described above, p38α can lead to the inactivation of TAK1 via the phosphorylation of TAB1 as the feedback control [13]. Furthermore, Miura et al. demonstrated that p38 can suppress JNK activity in a cross-inhibitory manner by inducing DUSP1 expression, in addition to a TAB1/TAK1 feedback loop [119]. Notably, JNK activity in response to inflammatory cytokines and stress conditions (UV and anisomycin) was significantly increased when several cell lines were pretreated with p38 inhibitors. If this phenomenon is universally observed in intact cells, we would have to say that it is an uninvited guest. In addition, the diversity of each SAPK-regulated function depends on its own many substrates and may produce unclear or undesired effects in the clinical trials using the SAPK inhibitor [4,9]. As one of the policies to solve this problem, an inhibition of the downstream branch/substrate of the SAPK isoform is promising. In fact, ATI-450 targeting the p38α–MK2 complex was shown to exhibit an anti-inflammatory effect without serious adverse effects in a phase I trial, although a target disease of ATI-450 in the current clinical study is COVID-19, not pulmonary fibrosis [98] (clinicaltrials.gov, registration number NCT04481685). Very recently, to precisely understand the pathogenesis of IPF, a single cell-based transcriptome analysis was in an active state [120,121]. Accumulating single cell-based informatics will reveal the exact spatial and temporal activation of SAPKs, which positively or negatively contribute to the functional outcome after IPF. As a result, an optimally programed SAPK inhibition may be a therapeutic option for IPF.

## Figures and Tables

**Figure 1 ijms-22-06041-f001:**
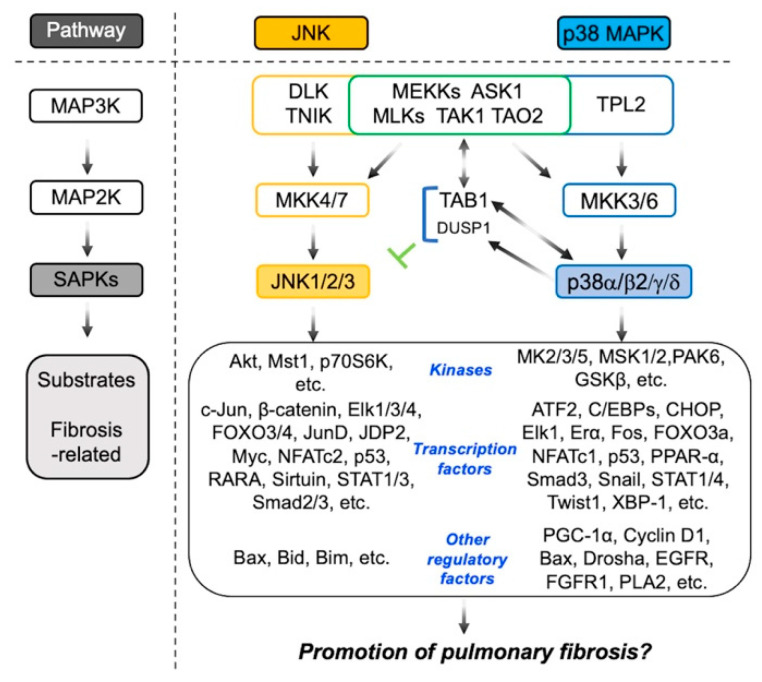
Schematic simplified representation of the JNK and p38 MAPK pathways in mammals. Both signaling pathways can be induced by inflammatory cytokines and profibrotic growth factors through an upstream activator sequence: MAP3K, MAP2K, and MAPK, promoting various regulatory proteins (kinases, transcription factors, transcriptional coregulators, post-transcriptional factors, and apoptosis-related factors) that are involved in the inflammatory and fibrotic responses.

**Figure 2 ijms-22-06041-f002:**
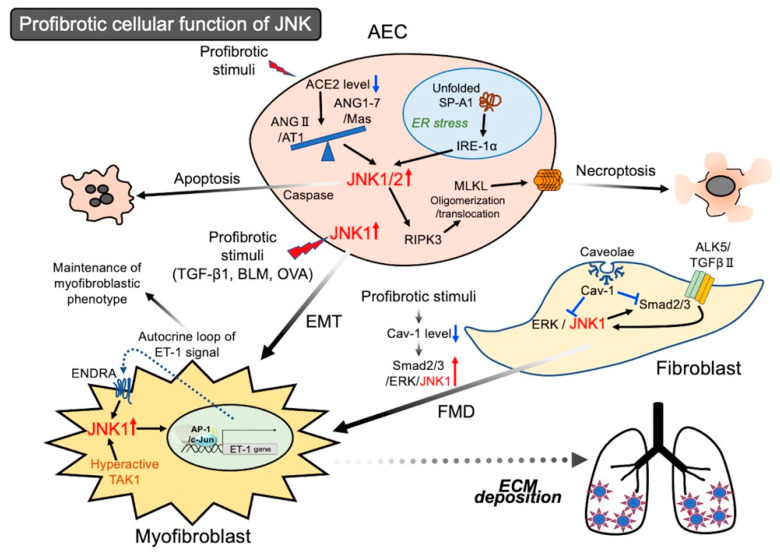
The molecular mechanism of JNK-mediated pulmonary fibrosis. JNK participates in various cellular responses to profibrotic stimuli: apoptosis, necroptosis, and EMT of alveolar epithelial cells (AEC); FMD of fibroblasts; and maintenance of the myofibroblastic phenotype.

**Figure 3 ijms-22-06041-f003:**
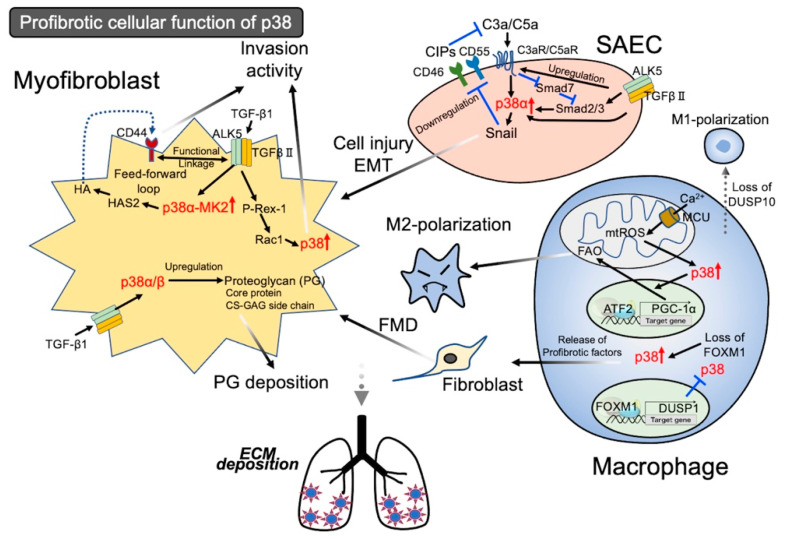
The molecular mechanism of p38-mediated pulmonary fibrosis. p38 participates in various cellular responses to profibrotic stimuli: EMT and cell injury of small airway epithelial cell (SAEC), invasion capacity and PG deposition of a myofibroblast, and M2 polarization and FMD promotion of a macrophage. Loss of DUSP10 leading to p38 activation seems to mediate the antifibrotic cellular functions. A more detailed evaluation regarding this point remains to be elucidated.

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
