# Peer review of "Pathophysiological Roles of Stress-Activated Protein Kinases in Pulmonary Fibrosis"

_ijms, 2021, doi:10.3390/ijms22116041_

Round 1

Reviewer 1 Report

The manuscript is well-written and snugly describes the role of SAPKs in molecular mechanisms of IPF. However, some improvements need to be made. 

The last paragraph of the manuscript about the anti-SAPK strategy should be increased a little bit. There are some more potential approaches to modulate stress-activated protein kinases in pulmonary fibrosis, for example via HSP90 inhibitors,  since they  modulate the activation of MAPK and JNK 

Author Response

As you pointed out, HSP90 is a very important mediator in the pathogenesis of IPF. Although HSP90 signal is mainly mediated by ERK, its correlation with SAPK signal is also shown and of interest. According to your suggestion, we incorporated new sentences (written in green) with new references into the last section (on lines 560-580 and 596-602).

Reviewer 2 Report

The review by Kasuya et al aims to summarize the current knowledge regarding the role of Stress-activated protein kinases (SAPK) in pulmonary fibrosis. The current review does not present the available knowledge in an easy to understand manner. Major concenrns:

19 The design of the manuscript must be improved. It might be better to present each SAPK in the context of different forms of pulmonary fibrosis (Idiopathic Pulmonary Fibrosis, Non-Specific Interstitial Pneumonitis, Cryptogenic Organizing Pneumonia, Lymphocytic Interstitial Pneumonitis, Respiratory Bronchiolitis associated Interstitial Lung Disease, Desquamative Interstitial Pneumonitis, Acute Interstitial Pneumonia). Alternatively, sub-divide each SAPK chapter by the different fibrosis pheno/endo-types.

2) The manuscript is often loosing its focus in to many details, that are either not well linked to the title or are difficult to be understood by any reader who is not very familiar witht he function of SAPK.

Some examples: locgically, the text in line 57-65 should be presented after line 50.

line 113: the SAPK described 20 years ago should be named.

lines 145-163: There is no link of the ACE-angiotensin system to SAPK? Not one SAPK is mentioned.  The same applies to lines 182-191.lines 222- 240, lines 322-348 and lines 350-364.

3) in many places the information on SAPK isoforms is missing, evenso the authors mentioned earlier that different isoforms or splicing forms exist. This applies especially to p38 MAPK alpha-gamma) chapter 3.2.2. If the cited publications do not mention which isoforms were investigated then it has to be stated and the lack of infromation needs to be discussed.  

4) The use of English terminology and phrasing needs to be corrected in many places e.g.: line 22-23 it is unclear what the pathological function of their lung cellular components" ,means. Is this about different cell layers? Or about different cell types? Or of lung compartments? Furthermore, instead of "regulation of the pathological fundtion" should be rephrased. This is better described as the impaired physiologic function. 

5) It would be nice if each chapter is summarized in a graphic overview rather then a very generic single graphics.

Author Response

The review by Kasuya et al aims to summarize the current knowledge regarding the role of Stress-activated protein kinases (SAPK) in pulmonary fibrosis. The current review does not present the available knowledge in an easy to understand manner.

[Response]

We appreciate your detailed suggestion.

Please note that newly incorporated sentences into the revised version in accord with your suggestion are written in red.

Major concerns:

1) The design of the manuscript must be improved. It might be better to present each SAPK in the context of different forms of pulmonary fibrosis (Idiopathic Pulmonary Fibrosis, Non-Specific Interstitial Pneumonitis, Cryptogenic Organizing Pneumonia, Lymphocytic Interstitial Pneumonitis, Respiratory Bronchiolitis associated Interstitial Lung Disease, Desquamative Interstitial Pneumonitis, Acute Interstitial Pneumonia). Alternatively, sub-divide each SAPK chapter by the different fibrosis pheno/endo-types.

[Response]

We mainly provide the overview of the association of SAPK with progressive pulmonary fibrosis to evaluate whether SAPK could be the potential therapeutic target. We have rewritten the fourth paragraph in Introduction section (on lines 125-135) to be more in line with your comments.

On the other hand, the pathophysiological mechanisms of Cryptogenic Organizing Pneumonia, Lymphocytic Interstitial Pneumonitis, Respiratory Bronchiolitis associated Interstitial Lung Disease, Desquamative Interstitial Pneumonitis, and Acute Interstitial Pneumonia, are distinctly different from those of progressive pulmonary fibrosis. We think that these phenotypes of Idiopathic Interstitial Pneumonia lacking progressive pulmonary fibrosis would be outside of our paper.

2) The manuscript is often losing its focus in many details, that are either not well linked to the title or are difficult to be understood by any reader who is not very familiar with the function of SAPK.

Some examples: logically, the text in line 57-65 should be presented after line 50.

[Response]

According to your suggestion, the sentences were moved to the appropriate paragraph (on lines 66-74).

Line 113: the SAPK described 20 years ago should be named.

[Response]

According to your suggestion, we rewrote the sentences with a new reference (on lines 155-156).

Lines 145-163: There is no link of the ACE-angiotensin system to SAPK? Not one SAPK is mentioned. The same applies to lines 182-191, lines 222-240, lines 322-348, and lines 350-364.

[Response]

Also related to next suggestion, we added all necessary information with new references in the revised version as follows: substrates of JNK (on lines 53-66); substrates of p38 (on lines 80-94); correlation between RA system and JNK isoform (on lines 198-203); correlation between ER stress and SAPK isoform (on lines 237-239); correlation between ET signal and JNK isoform (on lines 385-294); correlation between anaphylatoxins and p38 isoform (on lines 362-365); detailed signaling relationships between Smad and p38 in the case of proteoglycan production (on lines 390-393); correlation between MK2 and p38 isoform (on lines 426-431); comment regarding p38 isoforms (on lines 446-447); correlation between PGC-1α and p38 isoform (on lines 479-486); and discussion regarding the relationships between downregulation of DUSPS and SAPK isoform (on lines 503-508 and 514-517).

3) In many places the information on SAPK isoforms is missing, even so the authors mentioned earlier that different isoforms or splicing forms exist. This applies especially to p38 MAPK alpha-gamma (chapter 3.2.2.). If the cited publications do not mention which isoforms were investigated then it has to be stated and the lack of information needs to be discussed. 

[Response]

As mentioned above, we added the information regarding SAPK isoform candidate. Likewise, we mentioned how to define SAPK isoform in the overviews (on lines 321-329 for new Fig. 2 and lines 518-521 for new Fig. 3).

4) The use of English terminology and phrasing needs to be corrected in many places e.g.: line 22-23 it is unclear what the pathological function of their lung cellular components" means. Is this about different cell layers? Or about different cell types? Or of lung compartments? Furthermore, instead of "regulation of the pathological fundtion" should be rephrased. This is better described as the impaired physiologic function.

[Response]

According to your suggestion, we changed the sentence in the Abstract section (on lines 23-25).

5) It would be nice if each chapter is summarized in a graphic overview rather than a very generic single graphics.

[Response]

According to your suggestion, we changed figure1 and newly added Figures 2 and 3 in the Revised version.

Round 2

Reviewer 2 Report

Thank you for adding the requested details.